# Ultra-Sensitive Affordable Cementitious Composite with High Mechanical and Microstructural Performances by Hybrid CNT/GNP

**DOI:** 10.3390/ma13163484

**Published:** 2020-08-07

**Authors:** Mohammadmahdi Abedi, Raul Fangueiro, António Gomes Correia

**Affiliations:** 1Institute for Sustainability and Innovation in Structural Engineering, School of Engineering, University of Minho, 4800-058 Guimarães, Portugal; mohammadmehdi.abedi@gmail.com; 2Department of Mechanical Engineering, University of Minho, Campus de Azurém, 4800-058 Guimarães, Portugal; rfangueiro@dem.uminho.pt; 3Centre for Textile Science and Technology, School of Engineering, University of Minho, 4800-058 Guimarães, Portugal

**Keywords:** hybrid CNT/GNP, self-sensing, piezoresistivity, cementitious composite, mechanical, microstructural, durability

## Abstract

In this paper a hybrid combination of carbon nanotubes (CNTs) and graphene nanoplatelets (GNPs) was used for developing cementitious self-sensing composite with high mechanical, microstructural and durability performances. The mixture of these two nanoparticles with different 1D and 2D geometrical shapes can reduce the percolation threshold to a certain amount which can avoid agglomeration formation and also reinforce the microstructure due to percolation and electron quantum tunneling amplification. In this route, different concentrations of CNT + GNP were dispersed by Pluronic F-127 and *tributyl phosphate* (TBP) with 3 h sonication at 40 °C and incorporated into the cementitious mortar. Mechanical, microstructural, and durability of the reinforced mortar were investigated by various tests in different hydration periods (7, 28, and 90 days). Additionally, the piezoresistivity behavior of specimens was also evaluated by the four-probe method under flexural and compression cyclic loading. Results demonstrated that hybrid CNT + GNP can significantly improve mechanical and microstructural properties of cementitious composite by filler function, bridging cracks, and increasing hydration rate mechanisms. CNT + GNP intruded specimens also showed higher resistance against climatic cycle tests. Generally, the trend of all results demonstrates an optimal concentration of CNT (0.25%) + GNP (0.25%). Furthermore, increasing CNT + GNP concentration leads to sharp changes in electrical resistivity of reinforced specimens under small variation of strain achieving high gauge factor in both flexural and compression loading modes.

## 1. Introduction

Cementitious composites, as one of the most extensively used materials in the structure’s design and construction, are brittle and susceptible to cracking [1]. Damage and the failure of cementitious composite occur mainly over time due to numerous factors such as some inherent drawbacks, porosity, heterogeneity, or aggressive environmental conditions, rebar corrosion, or overexploitation, aging of materials, overloading and lack of maintenance over their service life [2]. Among the microstructure of hardened cementitious composites, there are many nanoscale porosities and cracks. These cracks are formed in the process of fabrication or exploitation [3]. As the nanoscale cracks grow and expand over time, larger micro-scale cracks can develop, which may cause structural failures [4,5,6]. Timely detection of these intrinsic damages in initial phases can prevent their progression and the sudden collapse of structures and greatly minimize potential consequences of deterioration in the service life of structures [7,8]. The process of continuously damage detection and evaluating the state of civil infrastructures which is commonly known as structural health monitoring (SHM), has attracted the attention of many researchers in the past decades. SHM systems are generally organized to catching real-time data of the structure through sensors that measure condition state’s parameters such as stress, displacement, strain, temperature, and etc. Despite recent advances, most SHM techniques involve a limited number of sensors which are distributed in large areas of structures [2]. Among all SHM methods, cement-based self-sensing composites with capabilities of intrinsic damage sensing based on the piezoresistivity principle, have provided a more sustainable, real-time and practical alternative to concrete structure damage detection [9,10,11]. Existence of the conductive phase is essential to achieve piezoresistivity in a cementitious composite. This conductive phase creates conductive electrical networks within the non-conductive cementitious matrix, based on percolation and quantum electron tunneling effects [12,13,14,15]. This conducting network is disturbed when the cementitious composites are subjected to damage or deformation which results in changing the electrical resistivity [16,17,18]. Sensitivity and performance of the cement-based self-sensing composites are extremely dependent on the type of conductive phase, their concentration as well as their distribution [9,19]. Although the various types of conductive fillers such as nickel powder, iron oxide nanoparticles, carbon black, carbon fiber, carbon nanofiber, graphene nanoplatelets (GNPs), and carbon nanotubes (CNTs) have been used as a conductive phase in the cementitious matrix, GNPs and CNTs have attracted more attention for developing multifunctional composite with high gauge factor and lower percolation threshold due to their unique electrical and mechanical properties [9,19]. Furthermore, reinforcing cementitious composite by highly dispersed CNTs and GNPs with low concentration can improve mechanical performance due to denser microstructural formation, increasing hydration rate and bridging mechanisms [20,21,22]. The optimal percentage of CNTs and GNPs which have been reported by researchers and lead to significant enhancement in cementitious composites mechanical performance were usually less than 0.3% by weight of cement. However, excessive ones cause porosities formation due to agglomeration of nanoparticles. [21,22,23,24]. Additionally, to achieve high-performance cement-based self-sensing composite with a high gauge factor, much greater CNT and GNP concentration (sometimes up to 30 times higher) is required [1,25,26,27]. Besides, the presence of a certain amount of nanoparticle agglomeration was reported as a crucial issue in significant increasing electrical conductivity as well as reducing the percolation threshold [28,29,30,31]. Therefore, highly sensitive and applicable cement-based self-sensing composites mostly have weakness microstructural behavior with low mechanical and durability performances.

Consequently, the piezoresistivity behavior of composites can have sharp changes and scatters in electrical resistance over time or at the end of each loading cycle due to the destruction of conductive networks. In this study, in order to overcome some of these drawbacks, a hybrid combination of CNT + GNP has been used as a conductive part in order to develop a novel high sensitive affordable cementitious composite with a low percolation threshold. The mixtures of these two nanoparticles with different 1D and 2D geometrical shapes can cause amplification of percolation and electron quantum tunneling which decreases the percolation threshold to a certain amount that can avoid agglomeration formation and reduce the production cost. Moreover, the synergic effects of CNTs and GNPs can improve mechanical, microstructural, and durability performances by filler function, bridging and/or deviation of cracks, and increasing hydration rate. As a first step of this achievement, an effective compatible and affordable technique for dispersion of hybrid CNT + GNP high concentration with Pluronic F-127 was proposed by Abedi et al. [32]. Using this technique different concentrations of CNT + GNP were dispersed and incorporated in a cementitious mortar. These composites were further characterized in terms of mechanical, microstructural, and durability by various test methods. Furthermore, the piezoresistivity behavior of the composite specimens was evaluated under compression and flexural cyclic loading tests.

## 2. Experimental Methods

### 2.1. Raw Materials and Characterization

Table 1 summarizes the characteristics of the GNPs and multi-wall carbon nanotubes MWCNTs as provided by the manufacturer [32]. The Morphology of GNP + CNTs (dry mix) was characterized using a scanning electron microscope as depicted in Figure 1.

For CNTs and GNPs dispersion in aqua suspension, a non-covalent surfactant (Pluronic F-127) was used in this study. Pluronic F-127 is a non-ionic triblock copolymer surfactant composed of two hydrophilic chains of polyoxyethylene (PEO) which placed in two sides of a central polyoxypropylene (PPO) hydrophobic chain. These side chains of hydrophilic PEO are similar to superplasticizers of polycarboxylate which are typically used in cementitious composites [33]. For this reason, Pluronic F-127 was found to be compatible with cementitious composites and could possibly improve its mechanical behavior and dry bulk density due to improved fluidity of the mortar. According to previous research, to prevent formation of any porosities caused by surfactant function, tributyl phosphate (TBP) with ½ of surfactant weight ratio was used as antifoam [23]. Their chemical structures are presented in Figure 2 [32]. The ordinary Portland cement type I (CEM I 42.5R) and CEN Standard sand (EN 196-1 and ISO 679: 2009) were used to prepare mortar mixtures. The chemical composition and grain size curve of cement and sand are presented in Table 2 and Table 3, and Figure 3 [32].

### 2.2. Carbon Nanotube (CNT) + Graphene Nanoplatelet (GNP) Dispersion Method

Nowadays, a successful feasible and compatible technique for hybrid CNT + GNP dispersion in aqueous suspension to be used in multifunctional cementitious composites has been developed by 10% Pluronic F-127 with the addition of TBP through 3 h sonication (80 w, 45 kHz) at 40 °C [32]. In this route, first TBP (50 wt.% of Pluronic) was completely dissolved in 225 mL water for 12 h using a magnetic stirrer at 800 rpm/min. Thereafter, 10% Pluronic (wt.% of carbon nanomaterials (CNMs)) added to the water and the suspension was mixed for 1 more hour by magnetic stirrer mixer. Further on, CNT + GNP was added and stirred continuously for 1 h. Afterward, suspensions were placed in a sonicator bath. A digital temperature regulator was used to adjust the temperature during the ultrasonication process by the circulation system through a radiator and sensors.

Under these mild mixing conditions, negligible structural damage is expected for the CNTs and GNPs. The Raman spectroscopy (Figure 4) was carried out on CNTs and GNPs using laser excitation with wavelength 532 nm to avoid adverse effects on CNMs structural quality such as edge-type defect, reduction of aspect ratio and sp2 domain crystallinity (La), that cause a deleterious influence on their mechanical and electrical properties [34,35].

### 2.3. Cementitious Composite Fabrication

Plain and CNT + GNP reinforced specimens were prepared through the mixing of prepared CNT + GNP aqua suspensions with ordinary Portland cement and standardized sand using a laboratory mixer following EN 196-1:1994 standard.

First, the required amount of cement (450 g) was poured into the mixer’s stainless steel bowl in order to prepare the mortar mixes. Then, the prepared CNT + GNP aqueous suspension with 225 mL water was added to cement (after re-measurement to ensure that water does not evaporate before and after the dispersion process) and the required amount of sand (1350 g) were poured into the mixing machine’s hopper.

The mixer was then run for 1.5 min, with the stainless steel blade’s rotational speed at 140 m/min, followed by a 30 s timeout and then run at a 285 m/min higher speed for another 2.5 min. Next, the mixture was shed into 160 mm × 40 mm × 40 mm prismatic molds and placed on a jolting machine for 1 min for vibrating compaction.

For 24 h the molds were placed in a humid atmosphere (99%) and then the samples were de-molded and kept underwater for 28 days of the hydration process. Four copper mesh in 40 mm × 50 mm dimensions, were embedded as electrodes in 40 and 60 mm distance from the middle of the specimens which were used for piezoresistivity behavior evaluation (Figure 5). The samples were identified by the variation of CNMs concentration in such a way that specimens GC (0.1%), GC (0.3%), GC (0.5%), GC (0.7%), and GC (1.0%) are contained 0.1%, 0.3%, 0.5%, 0.7% and 1.0% of CNT + GNP respectively (1:1 by weight of cement). For all specimens the content of cement and W/C are constant and equal to 33.3% by weight, and 0.5, respectively.

### 2.4. Mechanical and Microstructural Characterization

Flexural and compressive strength tests were carried out according to BS EN 196-1:1995 standard. In addition, apparent porosity and dry bulk density of the samples were measured as stated by ASTM C20 and BS EN 1015-10:1999 standards. The results are obtained by the mean of at least 3 specimens for flexural and 6 for compressive test according the test procedure. Compressive and flexural moduli at rupture moment (E_Cr_ and E_Fr_) were also calculated by Equations (1) and (2) respectively:(1)$$E_{Cr}=\frac{F_{c}L_{0}}{A\Delta L}$$
(2)$$E_{Fr}=\frac{F_{f}L^{3}}{4\delta wh^{3}}$$
where *F_c_* is the compressive load at rupture, *L*_0_ initial length of the specimen in the direction of loading, *A* is the section area of specimen perpendicular to the loading, Δ*L* is the *L*_0_ changes at rupture, *F_f_* is the flexural load at rupture, *L* is the length of the specimen, *δ* is the displacement of the specimen at rupture at the middle and under the loading axis, w and *h* are also width and height of the section. The fracture surface of specimens was characterized by scanning electron microscopy using an acceleration voltage of 10 kV and secondary mode of electron after coating with an Au–Pd thin film (30 nm) in a high-resolution sputter coater (Cressington 208HR) in order to investigate the microstructure. Furthermore, the ultrasonic test was performed for microstructural evaluation according to BS EN 12504-4 standard by ultrasonic test device through two probes along the longitudinal transverse axis. Moreover, specimens weight loss percentage was measured as a criterion of cementitious composite durability against freeze-thaw cycles. In this route, saturated specimens with similar dimensions were tested in the temperature range of −20 °C to 30 °C after 28 days of curing. The relative humidity of the chamber was 90% in 30 °C and the duration of each cycle was considered 12 h (Figure 6). Furthermore, relative dynamic modulus of elasticity were calculated by Equation (3) [36]:(3)$$RDM={\left(\frac{t_{s.0}}{t_{s.n}}\right)}^{2}$$
where $t_{s.n}$ is the transmit time after n cycles and $t_{s.0}$ is the initial transmit time.

### 2.5. Piezoresistivity Measurement

Specimens with embedded electrodes used for piezoresistivity tests were dried at 70 °C for 72 h after 90 days of curing to avoid moisture effect on electrical conductivity. In this study a four-probe method (Figure 4) by applying a direct current (DC) was used to evaluate cementitious composite piezoresistivity behavior under cyclic compression (10 KN by rate of 50 N·s^−1^) and flexural (500 N by rate of 2.5 N·s^−1^) loading. Electrical resistance was measured by two digital multimeter and one programmable power supply. One multimeter was connected to the power supply and to the outer electrode for measuring the intensity of current and the other was connected to the inner electrode for measuring of voltage difference. The electrical resistivity ρ(t) of each specimen was obtained from the average of five resistance measurements and was calculated by combining first and second Ohm’s low equations (Equations (4) and (5) respectively) as presented in Equation (6):(4)$$R=\frac{V}{I}$$
(5)$$R=\rho \frac{L}{A}$$
(6)$$\rho \left(t\right)=\frac{V\left(t\right)}{I\left(t\right)}\times \frac{A}{L}$$
where R is electrical resistance, I(t) is the current between outer electrodes, V(t) voltage difference between inner electrodes is the applied voltage, *L* is the spacing between the inner electrodes, *A* is the contact surface between electrode and composite. For the following assessment of cementitious composite piezoresistivity, the fractional change in resistivity (FCR) was calculated by Equation (7):(7)$$FCR=\frac{\rho \left(t\right)-\rho_{0}}{\rho_{0}}$$
where $\rho_{0}$ is the initial electrical resistivity which is measured before loading and ρ(t) is the resistivity at time t during the test. To evaluate the sensitivity of CNT + GNP reinforced specimens, the gauge factor (GF) is defined as the relative change in electrical resistivity over the strain (Equation (8)):(8)$$GF=\frac{FCR}{\varepsilon }$$
where ε is the applied strain along the axis of force and bottom of the specimen in compressive and flexural loading respectively.

## 3. Results and Discussion

### 3.1. Mechanical and Microstructural Cementitious Composite Characterization

Mechanical properties of the reinforced cementitious composite were evaluated by the flexural and compression tests. The results of these tests for cementitious composites reinforced by different CNT + GNP concentrations are shown in Figure 7 after 7, 28, and 90 days of hydration period as well as its corresponding coefficient of variation that has not exceeded 4%. According to the results, the presence of CNT + GNP among cementitious composite causes mechanical properties enhancement. Reinforcing cementitious composite by 0.1%, 0.3%, 0.5%, 0.7% and 1% concentration of CNT + GNP lead to increasing flexural strength by 12%, 31%, 37%, 23% and 25%, respectively, when compared to the plain mortar after 7 days of hydration period and 14%, 23%, 41%, 16% and 13% after 28 days.

The same analysis for compressive strength shows an increase of 13%, 21%, 28%, 20%, and 17% after 7 days and 8%, 14%, 36%, 23%, and 16% after 28 days of hydration period. The normalized results of flexural and compressive strength results for different CNMs concentration and hydration periods are presented in Figure 8. As can be seen, the general trend of results indicates optimal dosage by around 0.5% concentration of CNT + GNP achieving the best mechanical properties of the reinforced composite.

Flexural and compressive (rupture) modulus of plain mortar and CNM/mortar composites are provided in Figure 9.

The results indicated significantly higher flexural modulus of CNM-reinforced composites as compared to plain mortar. It is also clear that increasing CNMs concentration up to around 0.5% presented more improvement in flexural and compressive modulus as compared to plain mortar. The improvement in flexural and compressive modulus was quite high reaching up to 123% and 168% respectively in the case of 0.5% reinforced samples after 90 days of curing. Specimens failure modes under flexural and compressive loadings have been shown in Figure 10.

In fact, incorporating cementitious composite by CNT + GNP leads to increasing mechanical performance due to the bridging mechanism and reducing the porosities. However, an excessive increase of CNMs concentration can cause agglomeration and porosities formation which consequently can decrease flexural and compressive strengths [37]. Despite this decrease, it is noticed that the flexural and compressive strengths values of specimen GC (1.0%) are still greater than the value of plain mortar. These results demonstrate that the hybrid combination of these carbon nanoparticles CNT + GNP is more efficient for mechanical properties enhancement than reinforced cementitious composites that used individual CNMs (Table 4). This can be explained by the capacity of load carrying between CNMs and the adjacent cementitious matrix. Besides, GNPs can facilitate the CNTs dispersion which leads to an increase in overall strength [38]. Additionally, in nano intruded cementitious composite, crack propagation is stopped by various types of inclusions: pores, grains, fibers, aggregates, and particles [39]. CNTs and GNPs also block the cracks by deviating and/or arresting their propagating tips, like an obstacle (Figure 11).

### 3.2. Microstructural Investigation

The results of apparent porosities, dry bulk density, and ultrasonic wave passing time for different nano-intruded cementitious mortar are shown in Figure 12 and Figure 13, and Table 5.

These results show that incorporating CNT + GNP into the cementitious composite generally leads to reducing apparent porosities (denser microstructure) and consequently increases the density and decreases the ultrasonic wave passing time. The addition of 0.1%, 0.3%, 0.5%, 0.7%, and 1% CNT + GNP to the cementitious composite has decreased the apparent porosities by 2.1%, 4.0%, 5.2%, 3.4%, 1.6% after 7 days and 2.0%, 4.4%, 7.1%, 3.6% and 0.4% after 28 days of hydration period, respectively. It is also noticed an increase of dry bulk density about 2% by increasing CNT + GNP concentration to around 0.5%. However, by further increasing nanoparticles percentage, it will increase micro porosities resulting from GNP + CNT agglomerates formation in cementitious composite inducing in a decrease of the dry bulk density and mechanical properties of the composite.

A similar trend was also observed for ultrasonic wave passing time (Table 5). Indeed, increasing CNT + GNP to around 0.5% leads to reduced passing time, while increasing CNT + GNP more than 0.5% results in an increase of passing time. Furthermore, an increase of the curing age, e.g., of the increase of hydration products, results in a decrease of apparent porosity and an increase of dry bulk density with a decrease of ultrasonic waves passing time as a consequence of a denser microstructure.

### 3.3. Durability of CNT + GNP Reinforced Cementitious Mortar

Figure 14 shows the weight loss percentage of cementitious composites as a function of freeze–thaw cycles numbers after 180 cycles.

These results reveal the same trend of the previous test results with an optimal percent, around 0.5% of CNT + GNP concentration for the maximum resistance against climatic cycles. Moreover, the increase of hydration rate of reinforced specimens due to nucleation effects of CNMs leads to a decrease of the destruction of cementitious composite with the increase of the number of cycles. This is clearly observed by the decrease ratio of weight loss with the number of cycles in comparison with the results of the plain specimen.

The same tendency of the optimum around 0.5% of CNT + GNP concentration was also observed for the relative dynamic elasticity modulus after 180 freeze–thaw cycles as depicted in Figure 15. Incorporating 0.1%, 0.3%, 0.5%, 0.7%, and 1.0% CNT + GNP into the cementitious mortar increases the dynamic modulus by 5%, 19%, 23%, 17%, and 4%, respectively.

### 3.4. Piezoresistive Behavior

#### 3.4.1. Electrical Resistance Results

The results of electrical resistance of plain and CNT + GNP reinforced specimens in different hydration period are presented in Figure 16.

As can be noted, reinforcing cementitious composite by CNT + GNP leads to a significant decrease in electrical resistance. Incorporating 0.1%, 0.3%, 0.5%, 0.7%, and 1% CNT + GNP (1:1) into the cement mortar reduced electrical resistance 78%, 93%, 97%, 98%, and 99%, respectively after 7 days of hydration period compared to the plain specimen. These results after 90 days of curing slightly decrease for 69%, 85%, 95%, 96%, and 98%, respectively. It also shows that the percolation threshold for reinforced cementitious composite by CNT + GNP is probably between the concentration of 0.5% and 0.7%. Additionally, it can be observed that in all reinforced and plain cementitious composites, the electrical resistance increased by increasing curing age and hydration period. This is because of the formation of finer pores due to cement hydration which leads to a decrease in pore water and cuts off the conductive pathway of water [46]. Moreover, cement hydration process also can make a gap in CNT + GNP conductive paths by changing in cementitious composite microstructure and surrounding the CNMs as visible in Figure 17.

However, the increase of electrical resistance with curing time for the specimens reinforced with more than 0.5% concentration is lesser in comparison with other specimens, especially after 7 days of curing. The reason for this can be explained by the fact that the concentrations of CNT + GNP greater than the percolation threshold leads to the well-established continuous CNT + GNP conductive pathways which are not affected significantly by dried pores [46,47]. Table 6 summarizes results of other studies and shows the high efficiency of hybrid CNT + GNP used in this research. Indeed, the synergic effects of CNT + GNP create more electrical conductive paths by lower concentration and consequently reduces electrical resistivity significantly.

Figure 18 illustrate clearly that the combination of CNTs and GNPs with 1D and 2D geometrical shapes can increase the possibility of the formation of long-range connectivity in a random system.

The high specific surface area of GNPs can also increase the free surface for electron transmission by the quantum tunneling mechanism schematically illustrated in Figure 19.

#### 3.4.2. Cyclic Compression Test Results

Figure 20 shows the results of the fractional change in resistivity together with cyclic compression response for reinforced cementitious composite by different CNT + GNP concentration in function of time.

As can be seen, increasing the compression load leads to a decrease in the electrical resistivity by making conductive paths closer to each other and by contrast removing the load causes an increase in electrical resistivity.

The maximum fractional change in electrical resistivity of cementitious composite was also increased by increasing CNT + GNP concentration. In general, the fractional change in resistivity (FCR) value was negative under compression loading due to the decrease in electrical resistance during loading compared to the primary electrical resistance of the specimen which was measured before loading (Equation (7)). However, in all CNT + GNP concentrations, except 0.5%, after full load elimination the amount of specimen electrical resistance does not return to its original value. This can be a consequence of some internal defects (i.e., holes, native cracks, etc.) in the microstructure which are gradually reduced in the successive cycles of compression loading. In order to obtain more information, these results are also presented in terms of the axial strain as illustrated in Figure 21. It can be observed an increase of the strain level with the increase of number of loading cycles, except for 0.5% of CNT + GNP concentration. Moreover, the residual strain is matched with initial electrical resistivity at the end of each unloading.

The ratio of the compressive modulus at rupture (Ecr) to the compressive modulus at 10 KN loading (E_C10_) also has been shown in Figure 22.

As can be seen, the difference of modulus between rupture moment and at the top of each loading cycle (10 KN) for specimen CG 0.5% is greater compared to the rest. Hence, the residual strain of specimen CG 0.5% was lower at the end of each loading cycle.

The relationship between the FCR with strain is shown in Figure 23 for the different CNT + GNP concentrations. The results of adjustment of these relationships by power function regressions are also presented.

It can be observed that increasing CNMs concentration causes an increase in the fractional change in resistivity with an optimum sensitivity of the composite to strains around 0.7% of CNT + GNP concentration. Indeed, an excessive increase of CNT + GNP concentration reduces the sensitivity of the composite under compression loading due to specimen saturation of the conductive paths. It also appears that the relation between variation of strain and electrical resistance becomes more non-linear whatever the stiffness of the specimen is decreased by.

#### 3.4.3. Cyclic Flexural Test Results

Figure 24 and Figure 25 are the correspondent results for the flexural tests adopting the same analysis as for the compression tests. From Figure 24 it can be observed that contrarily to the compression tests the electrical resistivity increases with the flexural loading. This can be explained by the mixed mechanisms of compression and tensile in the beam section in conjunction with the susceptibility of the micro-cracking network created in the bottom of the section under tension. Therefore, conductive paths are cut off by these cracks resulting in an overall increase of the electrical resistivity under the increase of flexural loading

As for the compression tests, the FCR increases with the increase of CNT + GNP concentration and after unloading it is also observed that the electrical resistance does not return to its initial value, this being offset more significantly with the increase of the number of cycles (Figure 25). In contrast to compression loading mode, the FCR values were generally positive under flexural loading due to the increase in electrical resistance during loading compared to the primary electrical resistance of the specimen which was measured before loading (Equation (7)).

The ratio of the flexural modulus at rupture (E_Fr_) to the flexural modulus at 500 N loading (E_F500_) are shown in Figure 26. As can be observed, specimen CG 0.5% and CG 1.0% showed higher and lower stiffness respectively compared to the other specimens at the top of each loading cycle.

Concerning the relationship between the FCR with strain under flexural loading for the different CNT + GNP concentrations a large scatter is observed (Figure 27) when compared with the compression tests (Figure 23). It is noticed that increasing CNM concentration caused an increase in the slope of the FCR curve which shows more sensitivity.

#### 3.4.4. Gauge Factors

To better illustrate the strain sensing capabilities of CNT + GNP reinforced cementitious composite, the gauge factor for both types of flexural and compression cyclic loading is presented in Figure 28.

Table 7 shows that higher gauge factors can be achieved, giving enhanced sensitivity, for CNT + GNP reinforced cementitious composite for 0.5%, when compared with previous studies that used individual CNTs or GNPs. These results are very encouraging showing that the hybrid combination of these nanoparticles (CNT + GNP) is very efficient to achieve a sensitive self-sensing cementitious composite with good performance in terms of durability and mechanical performances.

## 4. Conclusions

In this study, the mechanical, microstructural, and durability properties of hybrid CNT + GNP reinforced cementitious mortar were evaluated in different hydration periods (7, 28, and 90 d). The sensitivity of specimens to the strain and stress were investigated under cyclic flexural and compression loading by measuring fractional changes in the electrical resistivity. The dispersion of nanoparticles (0.1%, 0.3%, 0.5%, 0.7%, and 1% CNT + GNP with equal proportions) was achieved by using Pluronic F-127 and TBP with 3 h sonication at 40 °C and following outcomes were obtained:An optimal concentration of CNT + GNP around 0.5% (1:1) shows the best performance in terms of durability (resistance against freeze–thaw cycles), microstructure and mechanical behaviour.Incorporating 0.5% CNT + GNP into the cementitious mortar led to increasing flexural strength by 37%, 41%, and 43% after 7, 28, and 90 days of curing respectively. These amounts for compressive strength were 28%, 36%, and 46% respectively.The improvement in flexural and compressive moduli was quite high reaching up to 123% and 168% respectively in the case of 0.5% reinforced samples after 90 days of curing.Scan electron microscopy, dry bulk density, apparent porosities, and ultrasonic wave passing time also showed the denser microstructure for reinforced mortar by 0.5% CNT + GNP.Evaluation of relative dynamic modules and weight loss of specimens after 180 freeze and thaw cycles also showed the best performance in terms of durability for the reinforced specimen by 0.5% CNT + GNP.Incorporating 0.5% CNT + GNP into the cementitious mortar led to the sharp change in electrical resistivity under cyclic loading which caused flexural and compression gauge factors by 398 and 460 respectively. This optimal percentage reveals significantly higher gauge factors when compared with the available results using individual nanoparticles.The results of this study provide proof of the concept that incorporating a low concentration of a hybrid combination of carbon nanotubes (CNTs) and graphene nanoplatelets (GNPs) in a cement mortar can provide self-sensing capabilities of the reinforced cementitious composite, enhancing microstructure, durability and mechanical performances.

## Figures and Tables

**Figure 1 materials-13-03484-f001:**
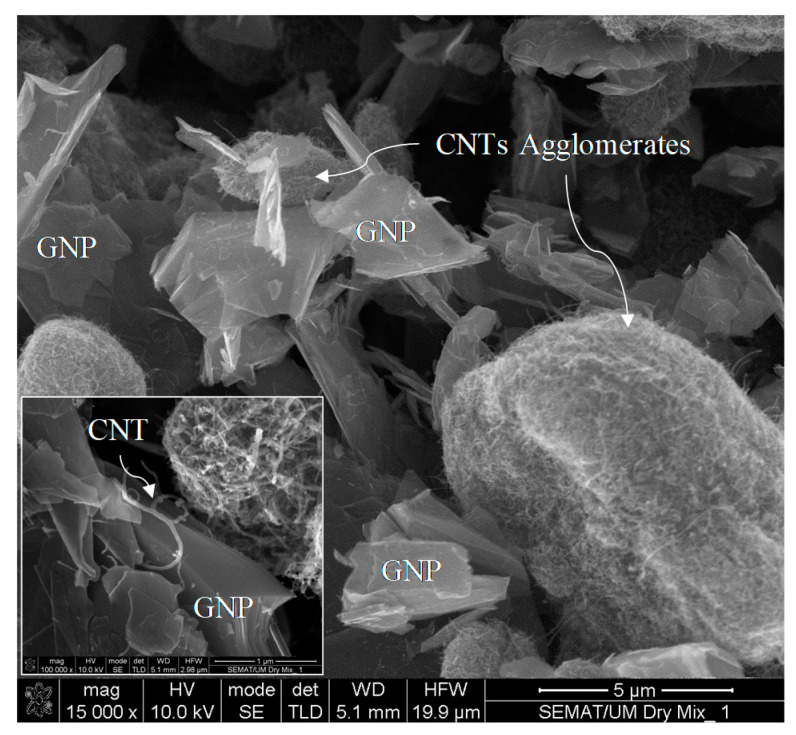
Morphology of carbon nanotubes (CNTs) and graphene nanoplatelets (GNPs) dry mixtures.

**Figure 2 materials-13-03484-f002:**
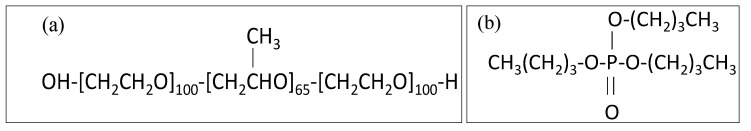
Chemical structure of: (**a**) Pluronic F-127, (**b**) tributyl phosphate 97% [32].

**Figure 3 materials-13-03484-f003:**
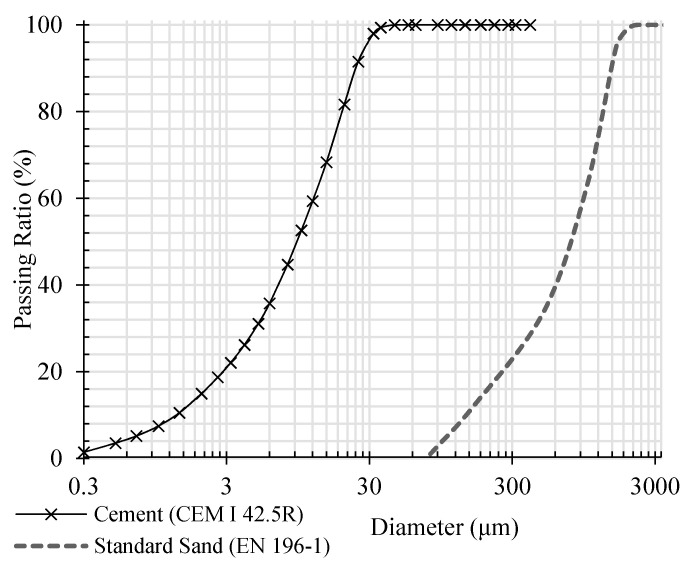
Size curves of cement and sand.

**Figure 4 materials-13-03484-f004:**
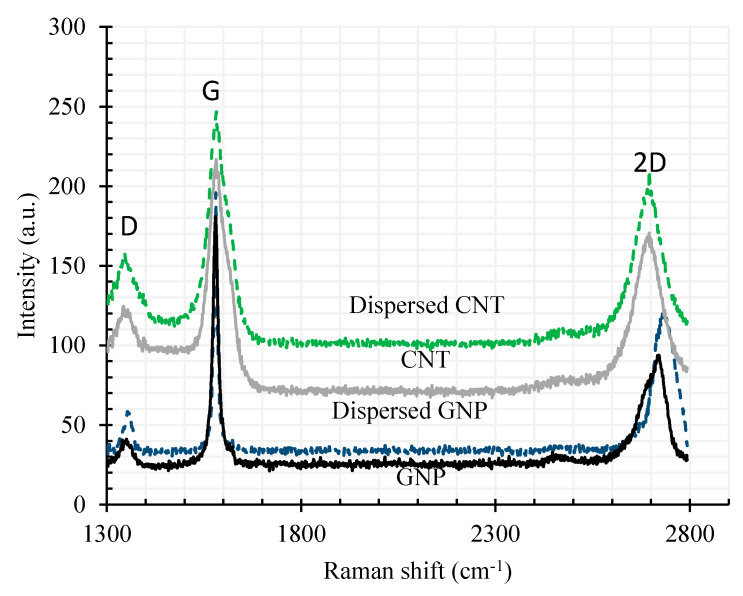
CNT and GNP Raman analysis results.

**Figure 5 materials-13-03484-f005:**
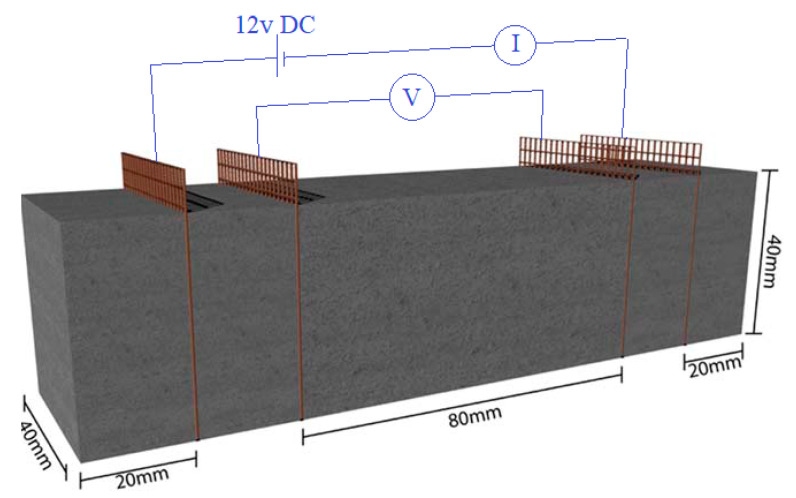
Representation of the geometrical parameters and resistivity measurement principle.

**Figure 6 materials-13-03484-f006:**
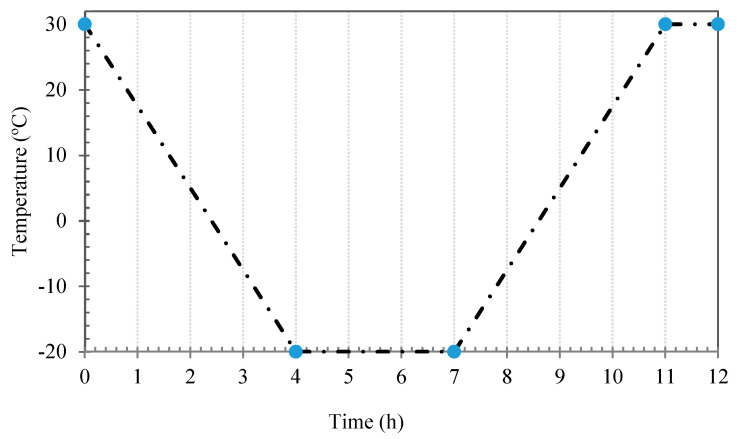
Freeze–thaw temperature cycle.

**Figure 7 materials-13-03484-f007:**
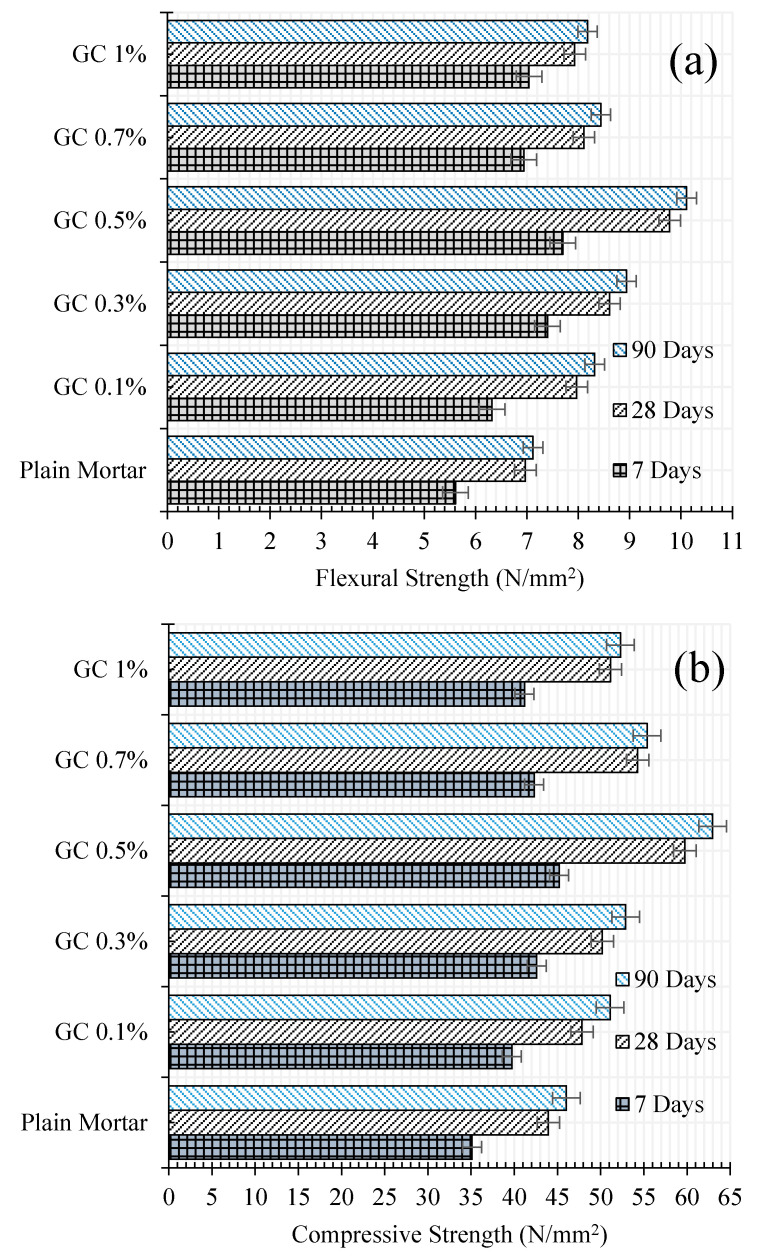
(**a**) Flexural and (**b**) compressive strengths of CNT + GNP incorporating cementitious composites at different curing time.

**Figure 8 materials-13-03484-f008:**
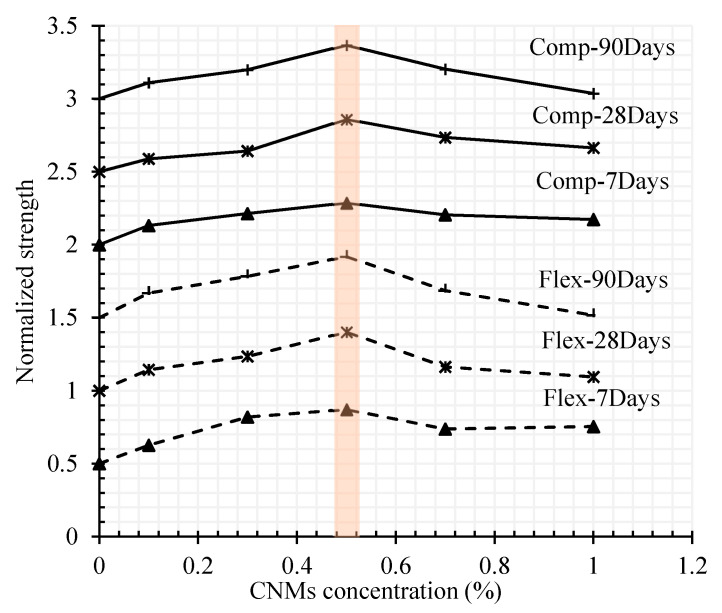
Normalized compressive (Comp) and flexural (Flex) strength of plain and CNMs reinforced mortar for different curing times.

**Figure 9 materials-13-03484-f009:**
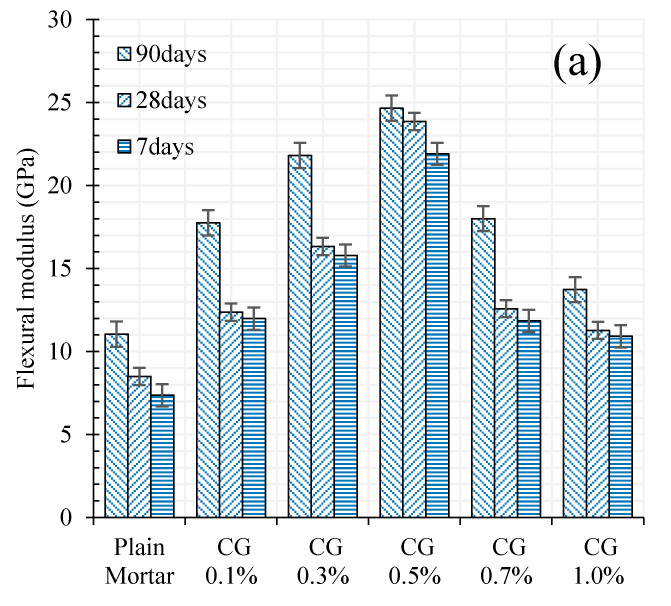
Plain and CNMs reinforced rupture modulus in different hydration periods: (**a**) flexural, (**b**) compressive.

**Figure 10 materials-13-03484-f010:**
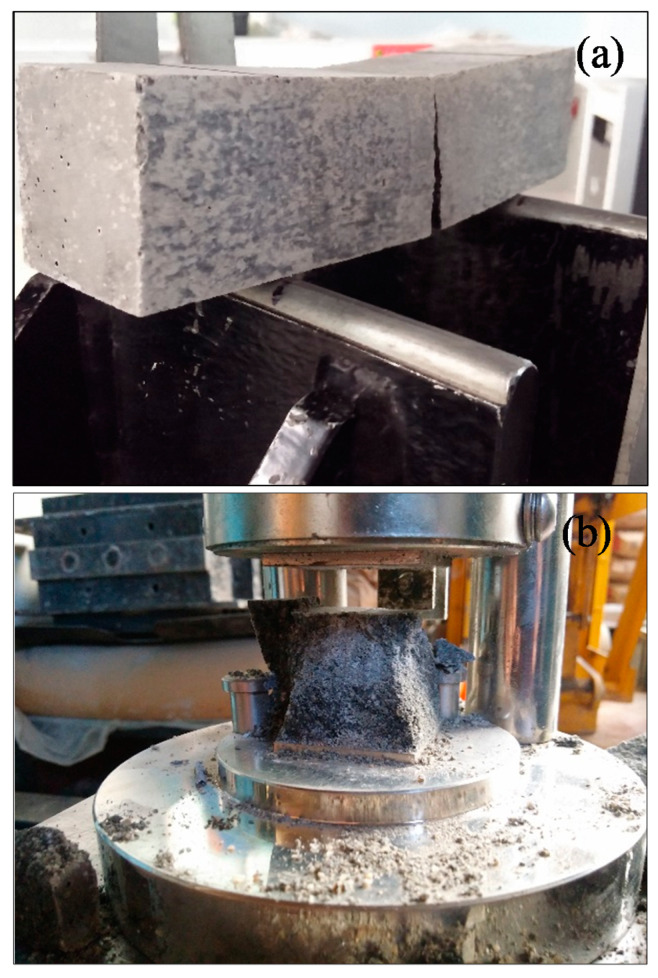
Images of specimens after tests: (**a**) specimen GC (0.5%) after flexural test, (**b**) specimen GC (0.5%) after compression test.

**Figure 11 materials-13-03484-f011:**
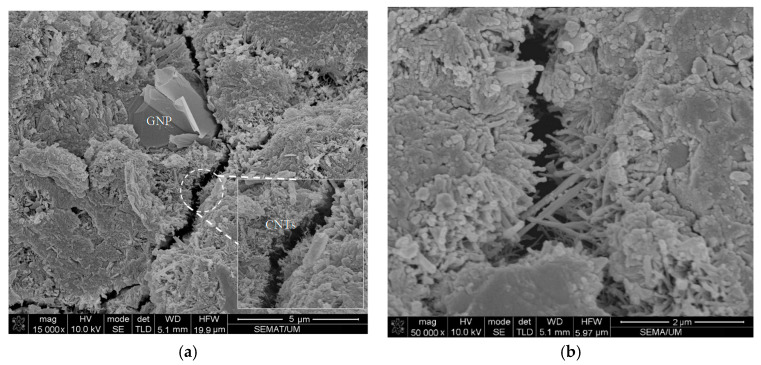
Scanning electron microscopy (SEM) morphology of CNT + GNP specimen after mechanical strength test. (**a**) GC (0.5%), (**b**) GC (0.7%).

**Figure 12 materials-13-03484-f012:**
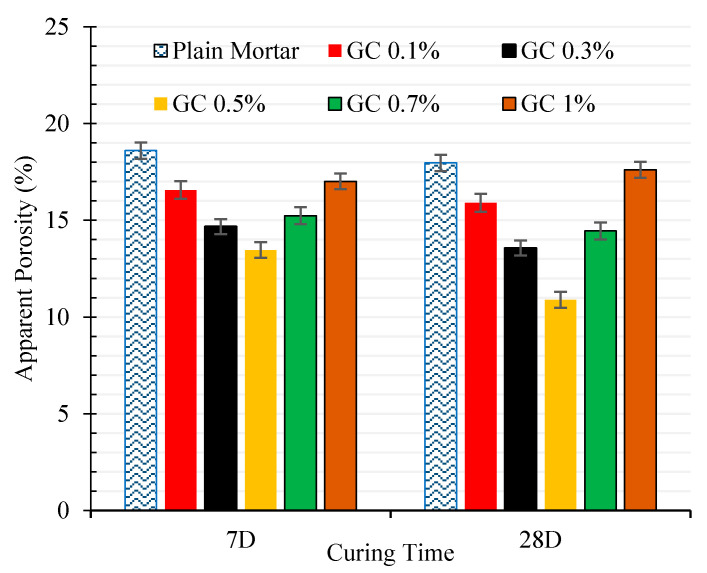
Apparent porosity of nano intruded cement mortars.

**Figure 13 materials-13-03484-f013:**
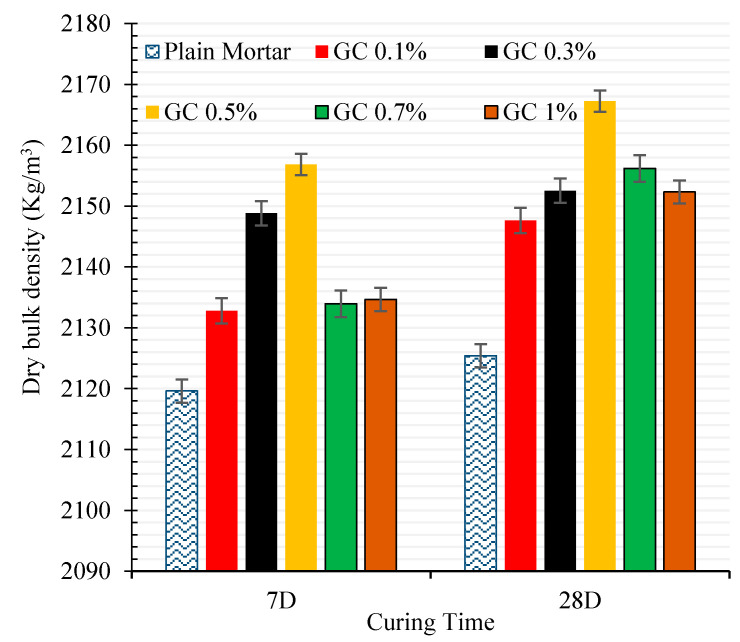
Dry bulk density of nano-intruded cement mortars.

**Figure 14 materials-13-03484-f014:**
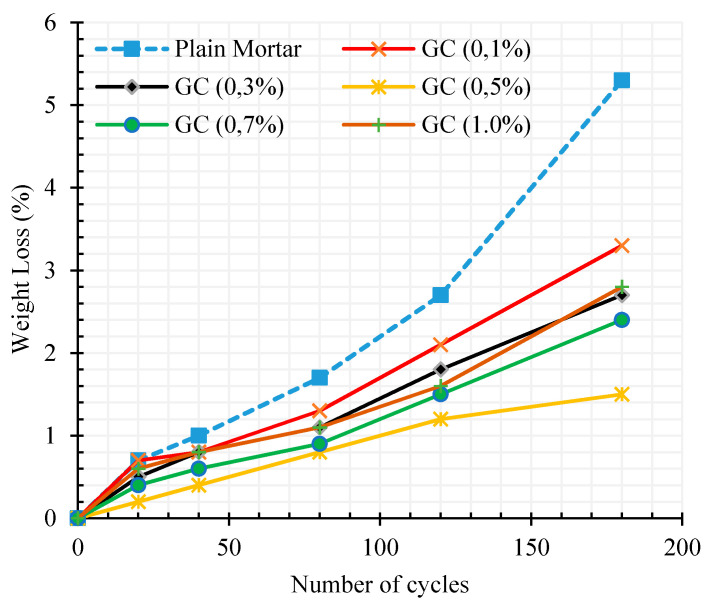
The weight loss rate of cementitious composites in different freeze–thaw cycles.

**Figure 15 materials-13-03484-f015:**
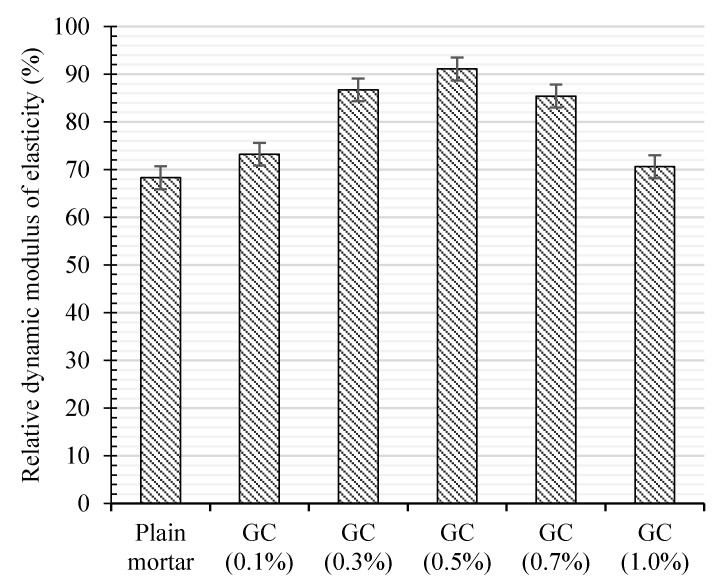
Relative dynamic elasticity modulus of CNT + GNP intruded cement mortar after 180 freeze-thaw cycles.

**Figure 16 materials-13-03484-f016:**
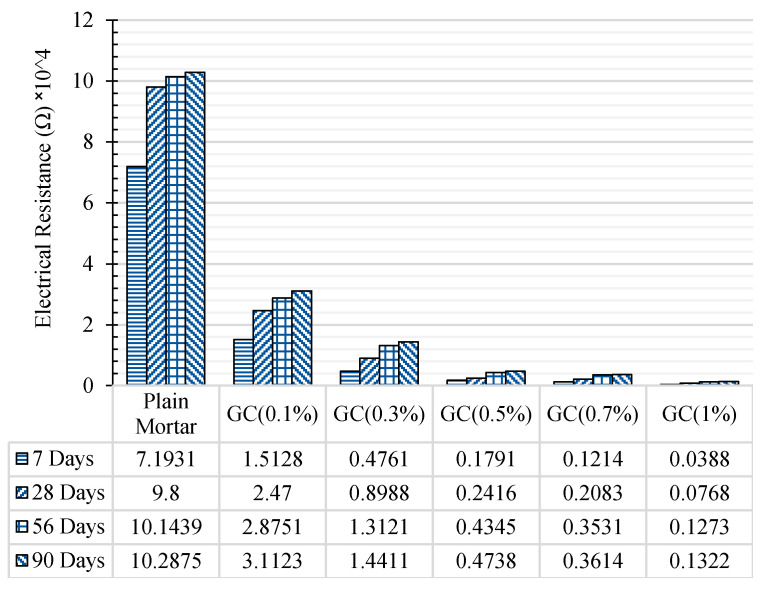
The electrical resistance of reinforced cementitious composite by different CNT + GNP concentrations and curing time.

**Figure 17 materials-13-03484-f017:**
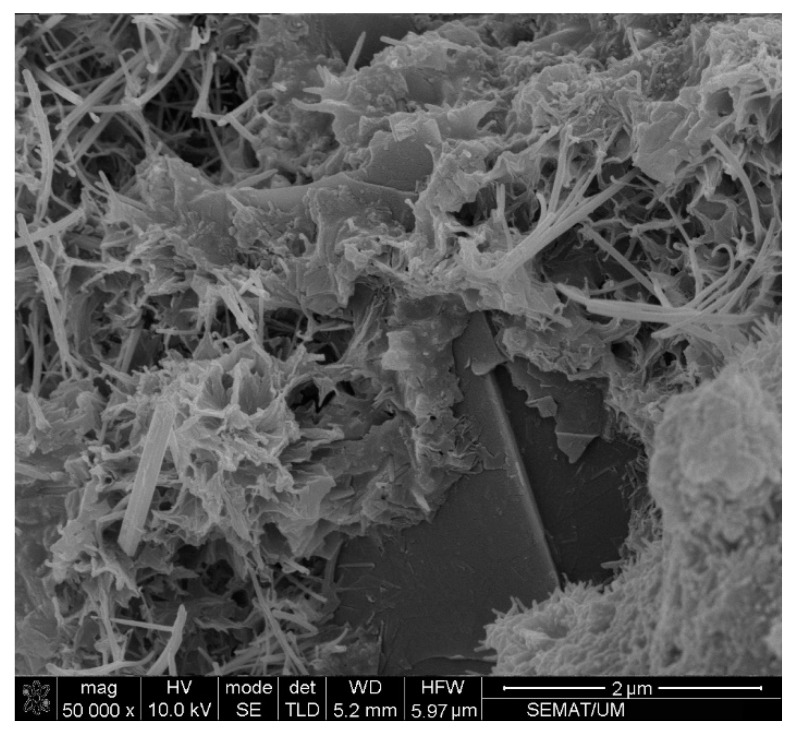
Buried GNPs and CNTs among of the hydration products.

**Figure 18 materials-13-03484-f018:**
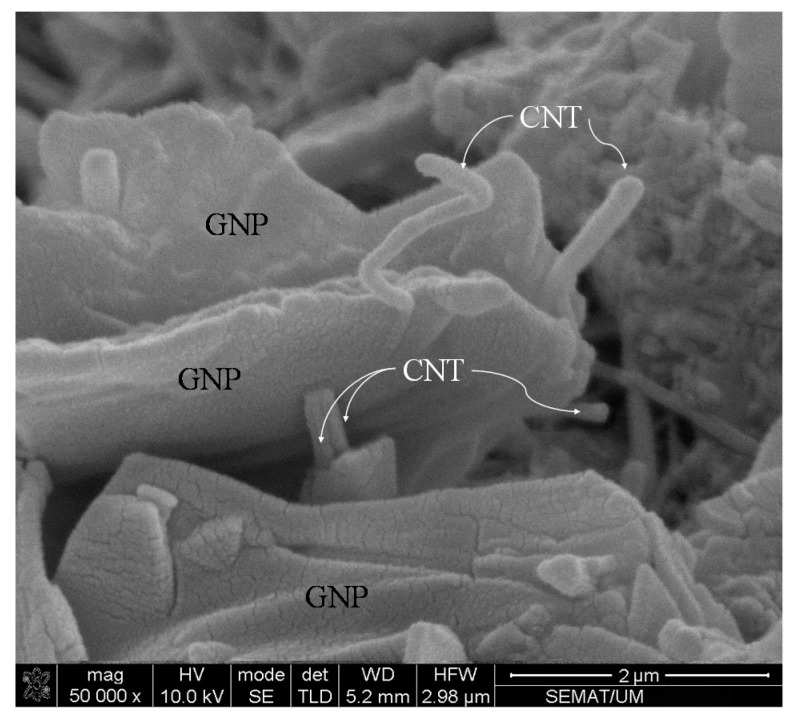
SEM image of incorporated CNT + GNP into the cement mortar.

**Figure 19 materials-13-03484-f019:**
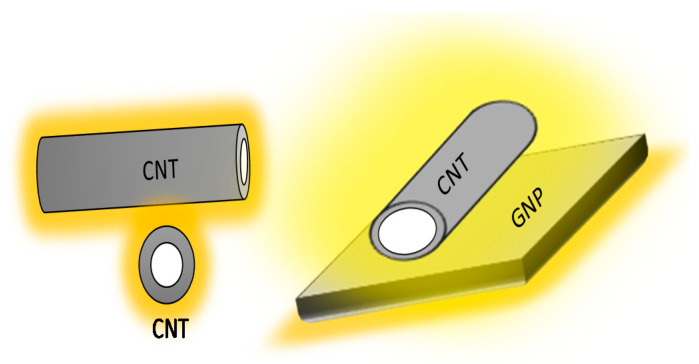
Schematic of tunneling effect for CNT/CNT and CNT/GNP cases.

**Figure 20 materials-13-03484-f020:**
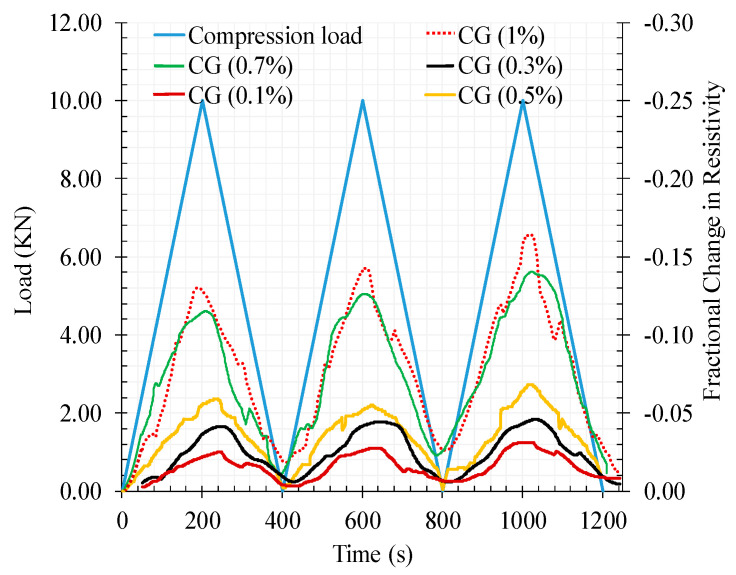
The fractional change in resistivity together with cyclic compression response for reinforced cementitious composite by different CNT + GNP concentration.

**Figure 21 materials-13-03484-f021:**
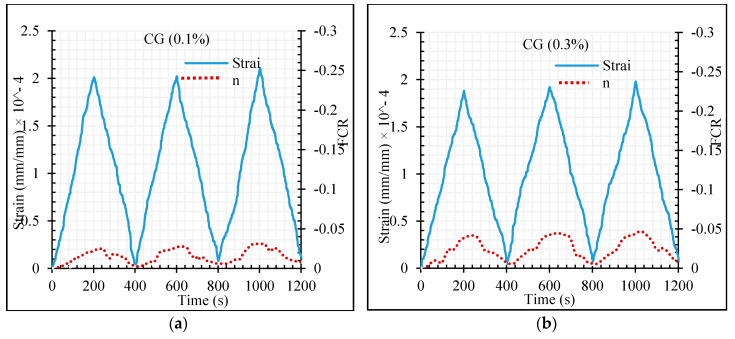
The fractional change in resistivity together with axial strain under cyclic compression loading. (**a**) GC (0.1%); (**b**) GC (0.3%); (**c**) GC (0.5%); (**d**) GC (0.7%); (**e**) GC (1.0%.

**Figure 22 materials-13-03484-f022:**
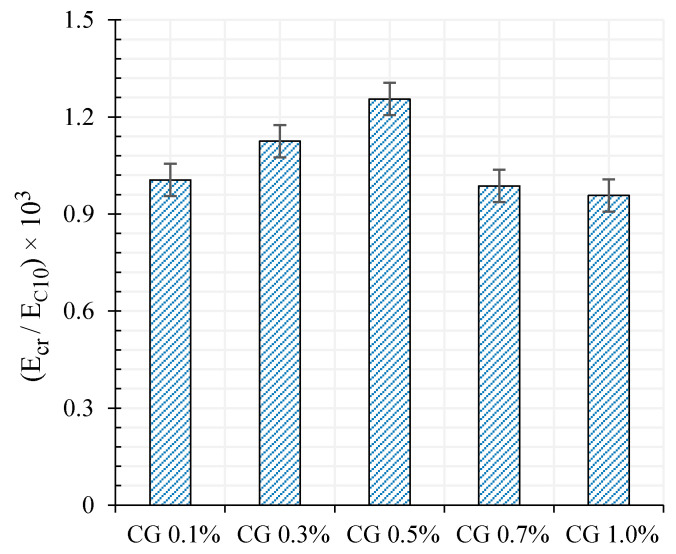
The ratio of rupture compressive modulus (Ecr) to compressive modulus at 10 KN loading (E_C10_).

**Figure 23 materials-13-03484-f023:**
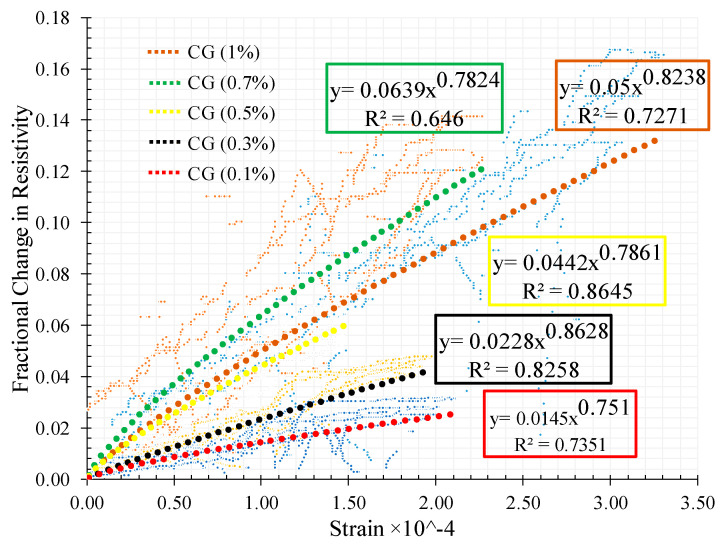
Variation of strain with the fractional change in resistivity for CNT + GNP reinforced specimens.

**Figure 24 materials-13-03484-f024:**
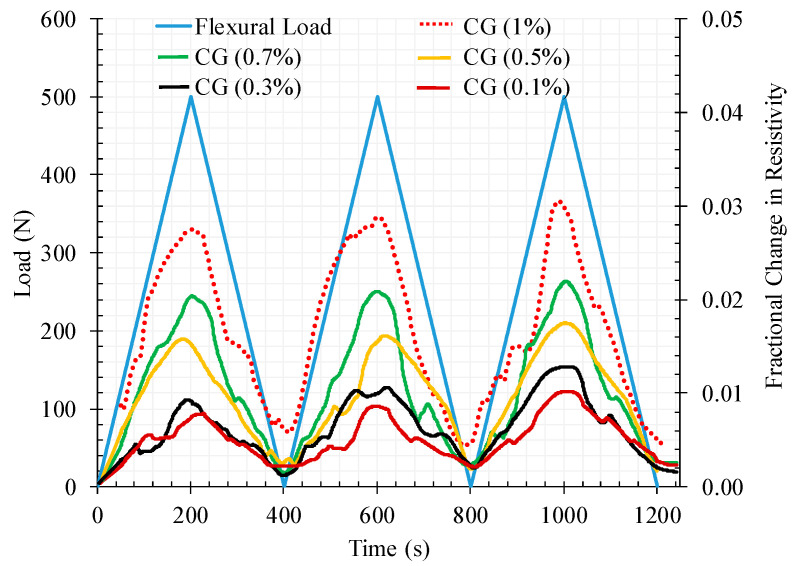
The fractional change in resistivity together with the cyclic flexural response for reinforced cementitious composite by different CNT + GNP concentrations.

**Figure 25 materials-13-03484-f025:**
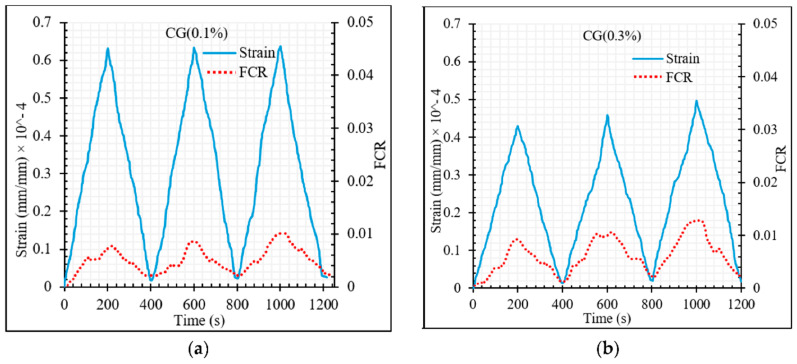
The fractional change in resistivity together with axial strain under cyclic compression loading. (**a**) GC (0.1%); (**b**) GC (0.3%); (**c**) GC (0.5%); (**d**) GC (0.7%); (**e**) GC (1.0%.

**Figure 26 materials-13-03484-f026:**
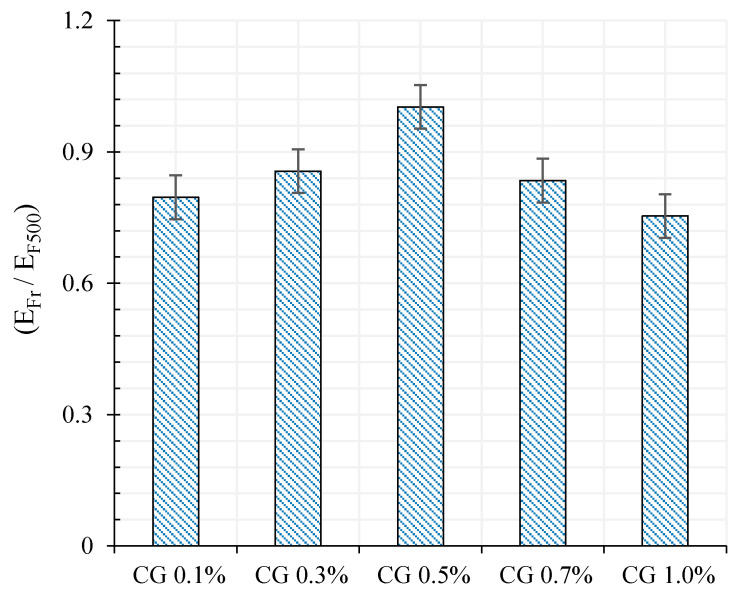
The ratio of flexural modulus at rupture (E_Fr_) to flexural modulus at 500 N loading (E_F500_).

**Figure 27 materials-13-03484-f027:**
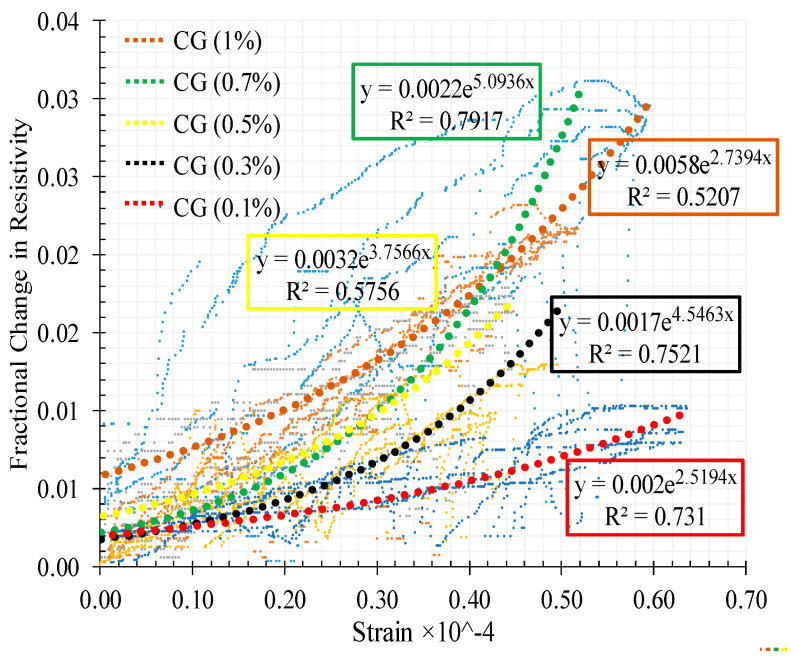
Variation of strain with the fractional change in resistivity for CNT+GNP reinforced specimens under flexural cyclic loading.

**Figure 28 materials-13-03484-f028:**
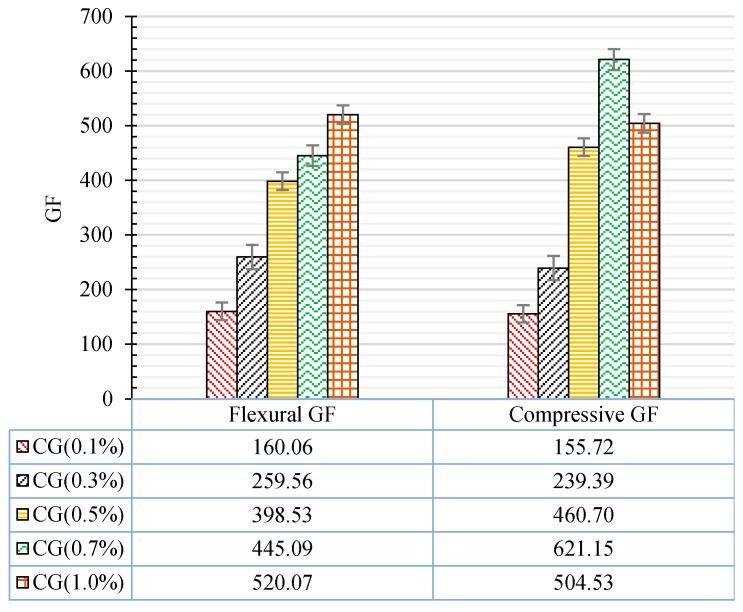
Variation of gauge factors in flexural and compression loading.

**Table 1 materials-13-03484-t001:** GNPs and CNTs characteristics.

**GNP**										
**Surface Area** **(m^2^ g^−1^)**	**Density (g/cm^3^)**	**Carbon Content (%)**	**Tensile Modulus (GPa)**	**PH Value (30 °C)**	**Tensile Strength (GPa)**	**Layers**	**Dimension**		**Form**	**Part Number**
120–150	0.6	>99.5	1000	7–7.65	5	<20	Thickness	Diameter	Gray Powder	TGN201
							4–20 nm	5–10 µm		
**MWCNT**										
**Surface Area** **(m^2^ g^–1^)**	**Density (g/cm^3^)**	**Color**	**Outside Diameter (nm)**	**Length (µm)**	**Ash (wt.%)**		**Carbon Content (%)**	**Part Number**		
350	0.27	Black	<8	30–10	<1.5		>98	GCM327		

**Table 2 materials-13-03484-t002:** Particle size distribution of the sand [32].

Mesh Size (mm)	0.08	0.16	0.5	1	1.6	2
Cumulative retained (%)	99 ± 1	87 ± 5	67 ± 5	33 ± 5	7 ± 5	0

**Table 3 materials-13-03484-t003:** Chemical composition and properties of ordinary Portland cement.

SiO_2_ (%)	Al_2_O_3_ (%)	Fe_2_O_3_ (%)	MgO (%)	CaO (%)	Na_2_O (%)	TiO_2_ (%)	K_2_O (%)	MnO (%)	P_2_O_5_ (%)	SO_3_ (%)	LOI ^a^ (%)	Finenes (m^2^/kg)	Specific Gravity (kg/dm^3^)	Initial Setting Time (min) ^b^	Soundness (mm) ^b^	Blaine’s Surface (cm^2^/g) ^b^
19.9	4.7	3.38	1.3	63.93	0.17	0.245	0.446	0.079	0.063	2.54	2.97	360	3.15	194	1.1	4220

^a^ (Loss on ignition)EN 196-2, ^b^ EN 196-3.

**Table 4 materials-13-03484-t004:** Comparison of mechanical properties with previous studies for CNMs reinforced cementitious mortar after 28 days of hydration.

Dispersion Method	Nanoparticles Type and Weight Fraction (% *)	Results and Improvement (%)	Standard	References
Ultrasonication	CNT (1%)	(6%) Compressive	N.A	[40]
Nitric and sulfuric acid by carboxylation plus Sonication	CNT (0.5%)	(19%) Compressive_(25%) Flexural	N.A	[41]
Modified acrylic polymer and acetone plus sonication and superplasticizers	CNT (0.5%)	(11%) Compressive_(33%) Flexural	UNI-EN 196-1	[42]
Modified polycarboxylate admixtures and naphthalene-sulfonate plasticizer	CNT (0.3%)	(12%) Compressive_(34%) Splitting Tensile_(14%) Modulus	NBR 7215, 8522,7222	[43]
Ultrasonication	CNT (0.5%)	(15%) Compressive_(36%) Splitting Tensile	ASTM C39 C496	[44]
Polycarboxylate superplasticizer	GO (1.5%)	(48%) Tensile	N.A	[45]
Pluronic F-127 and Tributyl phosphate by optimized temperature and ultrasonication	GNP + CNT (0.5% half by half)	(36%) Compressive_(41%) Flexural	BS EN 196-1:1995	Present Study

* by weigth of cement.

**Table 5 materials-13-03484-t005:** Ultrasonic wave passing times of nano intruded cement mortars.

Sample ID	Age	Ultrasonic Wave Time Passing (μs) for 150 kH	
		Longitudinal	Transverse
Plain Mortar	7D	36.46	10.69
	28D	36.40	9.60
GC 0.1%	7D	35.90	10.17
	28D	34.70	9.38
GC 0.3%	7D	35.20	9.61
	28D	34.10	8.93
GC 0.5%	7D	33.24	9.21
	28D	33.18	8.72
GC 0.7%	7D	33.70	9.43
	28D	32.90	8.64
GC 1%	7D	34.10	9.93
	28D	33.97	9.49

**Table 6 materials-13-03484-t006:** Comparison of electrical resistance for CNM reinforced cementitious composites after 28 days of curing.

Matrix	Nanoparticles Type	Weight Fraction (wt.%)	Electrical Resistivity (Ω·m)	References
Mortar	GNP	5	21.02	[1]
Paste	CNF	2	11	[16]
Mortar	GNP	5	78.2	[1]
Mortar	GNP	5	27.96	[1]
Mortar	CNT	0.7	24	[48]
Mortar	Graphene	1	4000	[49]
Mortar	Carbon Black	10	4.53	[50]
Paste	GNP	4.8	20	[3]
Mortar	Short Carbon fiber	2	2.4	[51]
Mortar	GNP + CNT	1 (half by half)	15.3	Present Study

**Table 7 materials-13-03484-t007:** Comparison of gauge factors for cementitious composite under compressive loading.

Matrix	Filler	Filler Ratio (Vol%)	GF	References
Cement Paste	Nanographite Plates	5	156	[26]
Mortar	GNP	2.4	30	[25]
Mortar	GNP	3.6	100	[25]
Mortar	GNP	2.1	90.8	[1]
Mortar	GNP	1.5	110	[1]
Mortar	CNT	0.15	240	[52]
Mortar	CNT + GNP	0.12	460	This study

## Data Availability

Requests for all types of data used to support the findings of this study, after the publication of this article, will be considered by the corresponding author, subject to obtaining permission from the owners.

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
