# Peer review of "Ultra-Sensitive Affordable Cementitious Composite with High Mechanical and Microstructural Performances by Hybrid CNT/GNP"

_materials, 2020, doi:10.3390/ma13163484_

Round 1

Reviewer 1 Report

The present study reports the results of an extensive experimental campaing aimed at investigating the influence of the several combinations of CNTs and GNPs for smart cementitous composites. The paper is well written and organized, the methods are rigorous and the topic is of interest for the literature and the Readers of the Journal.

Despite this, in this reviewer's opinion, some minor changes (more related to "formal aspects") are suggested before being accepted for publication.

As a general comment, the manuscript presents, in several parts unecessary figures/tables since most of the reported information are already described (or can be added)within the body of the text. In fact, the manuscript present 26 figures and 9 tables.

Specific comments:

  • Figure 1: please identify the CNTs and GNPs;
  • Table 4: maybe is not necessary. In fact, since most of the relevant parameters are kept constant (i.e., cement, water and sand amount), the mixtures composition can be described within the text;
  • Figure 7: not clear what the several colored lines represent;
  • Figure 8: please, describe better (in section 2) how the Flexural and compressive (rupture) modulus were calculated;
  • Figure 7 and Figure 12: use lines and not curves;
  • Figure 19 and Figure 23: improve quality of the images;
  • Figure 14 and Figure 26 use the same style adopted for the other figures (e.g., Figure 8).

Reviewer 2 Report

The article covers the topic of the Ultra-Sensitive Affordable Cementitious Composite with High Mechanical and
Microstructural Performances by Hybrid CNT/GNP. In my opinion, article presents valuable content. The subject and the supporting experiments are informative and present added value
to the body of knowledge on the subject area. The manuscript has acceptable cohesion.
However, some modification should be considered:

1. In case of cement mortar, constituent materials are very important and should be perfectly described.
Please add the grain size curve of cement and sand.
2. What type of cement was used in this research? Please provide cement class and technical description of this material. Please add the cement content.
3. I suggest to add point 2 - Research significance - Please descibe here the main essence of the research. (Why presented
paper is so important? What is major innovation in presented studies?).
4. Point 3.1. - How many specimens were tested in case of flexural and compression tests? Please add this information in the text and show the photo of specimens after tests.
5. Research has shown a great many interesting results that are insufficiently presented. I suggest that major conclusions should be presented point by point.

Author Response

Please see attachement.

Reviewer 3 Report

  1. Abstract -Abstract should be able to inspire subscriber interest. It is advisable to provide specific figures from the paper.
  2. Experimental Methods -What are the properties of the mortar mixed with Hybrid CNT/GNP? Was there any problem during the mixing process of mortar?
  3. Results and Discussion -In the experimental results, when the hybrid CNT/GNP is incorporated, the mortar's flexural strength and compression strength increase. Is there a reason In the experimental results, Hybrid CNT/GNP does not appear to react with cement.
  4. Experimental results -It is necessary to present specific figures in the conclusion. For excellence of manuscript, the presentation of specific data must be presented in the conclusion.
  5. Others -Please check the manuscript reference notation.

Author Response

Please see attachement.

Round 2

Reviewer 2 Report

All remarks have been considered by authors. Errors have been eliminated. The authors responded to all comments of the reviewer.
The current version is satisfactory for reviewer. In my opinion, article could be published.
Just a minor remark: The style of the article should be prepared in accordance to the rules in 'Materials' journal.

Reviewer 3 Report

All responses are completed and recommended.